# Platelet-Derived Procoagulant Microvesicles Are Elevated in Patients with Retinal Vein Occlusion (RVO)

**DOI:** 10.3390/jcm11175099

**Published:** 2022-08-30

**Authors:** Adrianna Marcinkowska, Nina Wolska, Boguslawa Luzak, Slawomir Cisiecki, Karol Marcinkowski, Marcin Rozalski

**Affiliations:** 1Department of Haemostasis and Haemostatic Disorders, Chair of Biomedical Sciences, Medical University of Lodz, Mazowiecka 6/8, 92-215 Lodz, Poland; 2Department of Ophthalmology, Karol Jonscher’s Municipal Medical Center, 93-113 Lodz, Poland; 3Platelet Signalling and Vascular Diseases, Institute for Clinical Chemistry and Laboratory Medicine, University Medical Center Hamburg-Eppendorf (UKE), 20246 Hamburg, Germany

**Keywords:** RVO, CRVO, BRVO, retinal vein occlusion, blood platelet, microvesicles, microparticles, thrombosis, eye, retina

## Abstract

The etiopathogenesis of retinal vein occlusion (RVO) is multifactorial, and the contribution of platelets to RVO development has not been fully elucidated. We aimed to analyze platelet function in RVO patients (*n* = 35) and controls (*n* = 35). We found a higher (*p* < 0.05) level of soluble P-selectin in RVO group vs. controls. Additionally, in RVO patients, the concentration of platelet-derived microvesicles was higher (*p* < 0.05), and the difference between groups was deeper for the fraction of platelet-derived microvesicles with the procoagulant phenotype (*p* < 0.0001) and for overall procoagulant microvesicles level (*p* < 0.0001). The results were similar for the total RVO group and for both RVO types (central- and branched-retinal vein occlusion). We did not find differences in simple platelet parameters (platelet count, mean platelet volume, platelet distribution width, platecrit, reticulated platelets) and inflammatory markers (platelet-lymphocyte ratio, neutrophil-lymphocyte ratio). Similarly, no differences were found for platelet aggregation-stimulated byadenosine diphosphate; collagen; arachidonic acid; and in multiparametric flow cytometry evaluation of P-selectin, PAC-1, and fibrinogen binding for both unstimulated and adenosine diphosphate-, collagen-, and thrombin receptor activating peptide-stimulated platelets. Our results suggest that platelets can contribute to developing RVO by enhancing procoagulant activity through providing a procoagulation surface via platelet-derived microvesicles. The direct role of platelets’ hyperreactivity in developing RVO is less apparent, which is consistent with the complexity and multifactorial background of this disorder.

## 1. Introduction

Retinal vein occlusion (RVO) is a retinal vascular disease characterized by the formation of a thrombus within the venous vessel, resulting in an outflow disturbance of blood from the retina [1]. The classification of this disorder is based on the site of the vessel occlusion. In consequence, two types of RVO exist—branch retinal vein occlusion (BRVO) and central retinal vein occlusion (CRVO) [1,2]. The prevalence of BRVO (0.4%) is higher than that of CRVO (0.08%) [1,3]. A similar frequency of the disease is observed in men and women, and in both cases, the risk of RVO increases with age [1,3]. The etiopathogenesis of retinal vein occlusion is believed to be multifactorial, and the contribution rates of various risk factors still need clarification [4]. In general, risk factors are both local, such as high intra-ocular pressure, glaucoma, or short axial distance, and systemic, involving hypertension, arteriosclerosis, hypercholesterolemia, diabetes mellitus, systemic vascular disease or inflammation, inherited thrombophilia, increased coagulability, age, obesity, smoking, and oral contraceptive use [2,5,6,7]. It should be noticed that among the above-mentioned systemic risk factors, hypertension is the most common systemic disease predisposing to the appearance of RVO [8,9]. It is estimated that arterial hypertension occurs in about two-thirds of RVO patients above the age of 50 years old [8]. Hyperlipidemia is also believed to be a significant risk factor of RVO and can be found in around one-third of patients with this disease [8]. Therefore, it is considered that arterial hypertension and hyperlipidemia are two independent factors that interacting together lead to compression of the arteriosclerotic altered retinal artery under the influence of elevated pressure on the retinal vein at an arteriovenous crossing point by sharing an adventitial sheath [10,11,12]. In light of the above considerations, it seems that the pathogenesis of RVO is a model example of Virchow’s triad, which assumes that a thrombosis is caused by changes such as vascular endothelial damage, turbulent blood flow, and concomitant abnormal hematological factors [10,12,13]. The manifestation of the described pathophysiological lesions on the fundus of the eye such as dilated, tortuous retinal veins; intraretinal hemorrhages; cotton wool spots; optic disc swelling; and macular edema [10].

Blood platelets play a significant role in the complex process of hemostasis [7]. Upon the loss of the integrity of endothelium, platelets adhere to uncovered collagen fibers and von Willebrand factor, then undergo an activation process that results in the release of biological agents from their granules and transform from a disc to a sphere shape with the growth pseudopodia [4,14,15,16]. In the last stage, platelets form aggregates [14]. Furthermore, non-adherent platelets delivering thrombin, ADP, and thromboxane A2 additionally strengthen platelet activation [16]. Importantly, larger platelets are characterized by the production of higher levels of thromboxane A2 and an increased expression of glycoprotein Ib and IIb/IIIa receptors, which is responsible for activation of platelets [17]. Despite the fact that arterial and venous thrombosis were considered to be separate phenomena for a long time, they are characterized by many common features in their pathophysiology as well as their risk factors [18,19]. Both types of thrombosis share the occurrence of a hypercoagulable state and inflammatory condition, incidence of prothrombotic microvesicles, and excessive platelet activation [18,19].

Phosphatidylserine is a phospholipid mainly distributed in the inner layer of the cell membrane. Upon cell activation or apoptosis, phosphatidylserine becomes exposed on the surface of the cells, providing a negatively charged procoagulant surface. facilitating binding coagulation factors. This process is associated with the release of small, membrane vesicles called microvesicles (MVs) of about 0.05–1.00 µm in diameter [20]. Many different types of cells including leukocytes, erythrocytes, endothelial cells, or smooth muscle cells release them from their membranes [20]. Importantly, a considerable amount, as much as 70–90%, come from blood platelets [20]. Platelet-derived MVs are ultra-micro membranous vesicles 0.1–1.0 µm in diameter, and they are also known as “platelet dust” [20,21]. There are many factors contributing to the release of platelet MVs from the platelet membrane, such as activation and apoptosis of platelets [20,22]. Platelet-derived microvesicles possess procoagulating properties associated mainly with the presence of phosphatidylserine, P-selectin, and platelets’ integrin receptors in their cell membranes [23]. Overproduction of microparticles is observed in numerous physiological and pathophysiological processes including hemostasis, thrombosis, inflammation, angiogenesis, apoptosis [24], and atherosclerosis [20]. More specifically, the elevated levels of platelet-derived vesicles were identified in a number of diseases characterized by a prothrombotic phenotype, such as coronary artery disease, acute coronary syndromes, venous thromboembolism, antiphospholipid syndrome, and thrombotic thrombocytopenic purpura or stroke [20,23,25]. Little is known so far regarding the role of various microvesicles and platelet-derived microvesicles in RVO.

There are a number of simple platelet parameters available in a standard laboratory blood count [7]. These parameters include platelet count (PLT); mean platelet volume (MPV), which is a marker of platelet size and activity [26]; platelet distribution width (PDW), which is an indicator of changes in platelet size [27]; plateletcrit (PCT), which indicates the quantitative abnormalities of platelets [27]; and reticulated platelets (RPs), which are young forms of platelets [28]. Regarding RVO, in the existing literature there are conflicting reports on the role of these platelet parameters.

As far as platelet reactivity is concerned, there are only a couple of papers reporting the increased platelet response to collagen or ADP [14], ADP [18], and thrombin [16]. These observations support the hypothesis that platelet hyperaggregability is an important factor contributing to the etiopathogenesis of retinal vein occlusion [13,14,16]. In the case of platelet microvesicles, we found only one study reporting that microvesicles originating from platelets, erythrocytes, leukocytes, and endothelial cells were elevated in patients with RVO [24].

It should be stressed that the pathogenesis of retinal vein occlusion is connected with a systemic and local inflammation process including a chronic inflammatory disorder, atherosclerosis [29,30]. There are numerous blood cells involved in the inflammatory reaction, such as neutrophils, macrophages, lymphocytes [29], monocytes [31,32] and platelets [30,33]. Interestingly, there are some reports demonstrating the association between RVO and inflammatory markers such as the platelet/lymphocyte ratio (PLR) [30,34,35,36] and the neutrophil/lymphocyte ratio (NLR) [30,34,36,37].

The primary aim of this study was to establish whether platelet morphology parameters are changed in patients with retinal vein occlusion. Secondly, we aimed also to compare the level of inflammatory markers (PLR, NLR) in RVO group vs. controls. In addition, we analyzed a wide set of platelet function parameters including ADP-, collagen- and arachidonic acid-induced platelet aggregation measured in whole blood; expression of P-selectin, PAC-1, and fibrinogen binding using flow cytometry; soluble markers of platelet activation in bloodstream (sCD62); and platelet-derived procoagulant microvesicles.

## 2. Materials and Methods

### 2.1. Patients

In total, 35 patients (11 females, 24 males, with a mean age 62.5 years) who were diagnosed with RVO between October 2020 and December 2021 participated in this study (Patients). These patients complained of sudden, painless, unilateral loss of vision. Exclusion criteria were an acute inflammatory state (CRP > 10 mg/mL), use of anti-platelet and anticoagulant drugs, glaucoma, or optic nerve disorders. The control group involved 35 sex- and age-matched healthy volunteers (13 females, 22 males, with a mean age 59.1 years), who were subjected to a routine ophthalmic examination. The characteristics of RVO patients are shown in Table 1. All the hypertension patients (40%) were on hypertensive drugs. Similarly, all the dyslipidaemia patients (60%) were on statin therapy (9 patients took atorvastatin, 3 patients rosuvaststin, and 1 simvastatin). A medical interview allowed us to obtain information associated with the duration of RVO symptoms; medications taken; and the existence of risk factors, such as hypertension, hypercholesterolemia, diabetes mellitus, and glaucoma. Similarly, the control group was examined relating to ocular complaints, medications taken, and risk factors. In all participants, including the control group, a complete ophthalmic examination using slit lamp biomicroscopy and fundoscope was carried out. The diagnosis of RVO was based on the presence on the fundus of the eye symptoms, such as dilated and tortuous retinal veins, intraretinal hemorrhages, cotton wool spots, optic disc swelling, and macular oedema; CRVO changes were recorded in four retinal quadrants, while BRVO signs were noted in the retina drained by a closed vein vessel. The study protocol was approved by the Bioethical Commission of Medical University of Lodz, approval No. RNN/15/20/KE. Each of the participants gave written informed consent to participate in the study.

### 2.2. Blood Collection

Blood was collected on 3.2% sodium citrate (final citrate:blood ratio of 1:9 *v*/*v*) for studies of platelet function and on EDTA for blood count. Blood samples were taken after a fasting period of at least 12 h. Platelet reactivity was measured in whole blood, whereas CD62 and MV levels were analyzed in platelet-free plasma (whole blood was first centrifuged at 2500× *g* for 15 min to obtain platelet-poor plasma; subsequently, the platelet-poor plasma was transferred to a new centrifugation tube and centrifuged at 10,000× *g* for 5 min to obtain essentially platelet-free plasma). The plasma samples were stored at −80 °C until measurements.

### 2.3. Parameters from Peripheral Blood Morpholog

Standard parameters of peripheral blood morphology were obtained from the Hospital Diagnostic Unit: RBC, red blood cells; WBC, white blood cells; NEU, neutrophils; EOS, eosinophils; BAS, basophils; LYM, lymphocytes; MON, monocytes. In addition, platelet-related parameters of peripheral blood morphology were obtained: PLT, platelet count; MPV, mean platelet volume; PCT, plateletcrit; PDW, platelet distribution width; RET, reticulated platelets.

### 2.4. Platelet Aggregometry in Whole Blood

The measurements were carried out using a Multiplate aggregometer (Roche, Basel, Switzerland) according to the manufacturer’s protocol. Blood samples (300 µL) were transferred into the measurement cells and diluted with 300 μL saline (0.9%) and preheated to 37 °C for 3 min. Subsequently, either ADP at 6.4 μM (final concentration), collagen at 3.2 μg/mL (final concentration), or arachidonic acid at 0.5 mM (final concentration) was added, and platelet aggregation was recorded continuously for 6 min using a Multiplate analyzer. The area under the curve (AUC) measured in arbitrary units*min represented a rate of platelet aggregation. All the measurements were performed within 2 h of blood collection.

### 2.5. Expression of P-Selectin and PAC-1 Measured by Flow Cytometry

Platelets were either unstimulated or activated with collagen (in the concentrations 5 and 20 µg/mL), ADP (in the concentrations 1 and 10 µM) and TRAP6 (in the concentrations 1 and 10 µM) for 5 min at RT. Samples were then diluted 10-fold with PBS, labelled with anti-CD61/PerCP, anti-CD62P/PE, and PAC-1/FITC antibodies (20 min, RT), and fixed with CellFix for 1 h at RT. Directly before measurement, the samples were diluted 1:1 with PBS and the assay was performed, gathering 10,000 CD61/PerCP-positive events, using a FACSCanto II flow cytometer (BD Bioscience, Franklin Lanes, NJ, USA). The percentage of marker-positive platelets (above isotype cut-off) was measured.

### 2.6. Binding of Exogenous Fibrinogen Using Flow Cytometry

Exogenous Oregon Green-labelled fibrinogen was added to the samples (3 µg/mL), which were subsequently either unstimulated or activated with collagen (in the concentrations 5 and 20 µg/mL), ADP (in the concentrations 1 and 10 µM) and TRAP (in the concentrations 1 and 10 µM) for 5 min at RT. Samples were then diluted 10-fold with PBS, labelled with anti-CD61/PE antibodies (20 min, RT), and fixed with CellFix (prepared according to manufacturer instructions) for 1 h at RT. Directly before measurement, the samples were diluted 1:1 with PBS and the assay was performed, gathering 10,000 CD61/PE-positive events, using a FACSCanto II flow cytometer (BD Bioscience, Franklin Lanes, NJ, USA). The percentage and marker-positive platelets as well as mean fluorescence intensity (MFI) were measured.

### 2.7. Microvesicles Determination

Plasma samples were stained with the following antibodies: CD41a-FITC (BD Pharmingen), annexinV-PE (BioLegend, San Diego, CA, USA), CD45-BrilliantViolet421 (BioLegend), and CD144-PE/Cy7 (BioLegend) (1 µL of each per 50 µL of plasma) for 15 min at 37 °C, then fixed with PFA and diluted with PBS directly before the measurement. PFA and PBS were filtered through a 0.1 µm filter to reduce the background. Samples were measured on a NovoCyte Quanteon Flow Cytometer (Agilent, Santa Clara, CA, USA), flow rate slow, with no gating, for a fixed amount of time. Obtained data are presented as either densities (events/µL) or median florescence intensity (MFI). A flow cytometry schema of gating platelet-derived and procoagulant microvesicles is shown in Figure 1A.

### 2.8. Soluble P-Selectin Assay

The concentration of plasma-soluble P-selectin (CD62s) in platelet-free plasma was assessed by dedicated ELISA tests (Human sP-selectin ELISA kit, Cat. No. BMS219-4; with a sensitivity of 0.20 ng/mL and an inter-assay coefficient of variation (CV%) of 5.4%) according to the manufacturer’s instructions (Invitrogen, Waltham, MA, USA).

### 2.9. Statistical Analysis

Statistical analysis was performed in Statistica 13.1 (Statsoft, Cracow, Poland) and GraphPad Prism 8 (San Diego, CA, USA). The normality of the distribution of the analyzed variables was assessed using the Shapiro–Wilk test. Since the majority of variables were not normally distributed, data in tables and figures are presented as median and interquartile range: Me (Q1; Q3). For normally distributed variables, the statistical significance of differences between two groups was estimated using the unpaired Student’s *t*-test; for variables departed from normality, the Mann–Whitney U test was applied. To compare differences between more than two groups, analysis of variances (one-way ANOVA) with Bonferroni’s multiple comparisons test was used for normally distributed variables. For variables departed from normality, to compare differences between more than two groups, the Kruskal–Wallis test with Dunn’s multiple comparison test was applied.

## 3. Results

Basic characteristics of RVO patients are shown in Table 1. For details, see Section 2.1 in Materials and Methods.

**Table 1 jcm-11-05099-t001:** Basic characteristics of RVO patients (*n* = 35).

Parameter	Number of Subjects (%)
Age [years]	62.5 ± 13.7
Sex	11 F (31%), 24 M (69%)
Type of RVO	CRVO 21 (60%) BRVO 13 (39.7%) Unclassified 1 (0.3%)
Hypertension	21 (60%)
Diabetes	9 (25.7%)
Dyslipidaemia	14 (40%)
Glaucoma	12 (34.3%)

BRVO—Branch Retinal Vein Occlusion, CRVO—Central Retinal Vein Occlusion, RVO—Retinal Vein Occlusion.

### 3.1. Parameters and Indexes Calculated from Peripheral Blood Morphology

We measured the basic parameters available in the peripheral blood morphology in RVO patients, including the CRVO and BRVO subgroups, as well as control subjects. We found significantly higher cell count values in RVO for WBC, NEU, EOS, BAS, LYM, and MON, but not RBC (fve 2). In both subgroups of RVO patients, statistically significant elevation of WBC, NEU, LYM, and MON was observed in CRVO patients vs. controls, whereas no changes were found for BRVO vs. controls (Table 2).

We also determined the platelet-related parameters from peripheral blood morphology, such as platelet count, mean platelet volume, plateletcrit, platelet distribution width, and reticulated platelets. We found no statistically significant differences between the RVO group or subgroups and controls (Table 3).

### 3.2. Inflammatory Markers

On the basis of peripheral blood morphology (blood count) parameters, we calculated inflammatory indexes: the platelet–leukocyte ratio (PLR) and neutrophils–lymphocytes ratio (NLR). A comparison of these markers between RVO patients, including CRVO and BRVO subpopulations, and the control group did not yield significant differences between the groups (Table 4).

### 3.3. Platelet Activation and Reactivity Markers, and Platelet Aggregation

We analyzed basal platelet activation ex vivo (without stimulation with agonists) using standard flow cytometric markers such as P-selectin (CD62) and PAC-1 measured in whole blood. Additionally, we assessed binding of exogenous fluorophore-labelled fibrinogen to platelets. No statistically significant differences were found between RVO patients, including CRVO and BRVO subgroups, as compared to controls (Appendix A).

We also used a model of measuring platelet reactivity in vitro, based on whole blood stimulation with platelet agonists. We applied the same markers of platelet activation as described above and used three platelet agonists: ADP, collagen, and TRAP (thrombin-activating peptide). Since our idea was to check platelet response to agonists under both mild and strong activation conditions, we used each agonist at two concentrations. ADP was used at concentrations of 1 and 10 µM, collagen at concentrations of 5 and 20 µg/mL, and TRAP at concentrations of 1 and 10 µg/mL. No statistically significant differences were found between RVO patients, with the subgroups of CRVO and BRVO, as compared to controls (Appendix A).

Platelet aggregation was measured in whole blood using the impedance method and a set of platelet agonists in concentrations recommended for this assay by the manufacturer (ADP 6.4 µM, collagen 3.2 µg/mL, and arachidonic acid 0.5 mM). No statistically significant differences were found between RVO patients, with the subgroups of CRVO and BRVO, in comparison to the control group (Appendix A).

### 3.4. Procoagulant and Platelet-Derived Microvesicles

We found that RVO patients were characterized by a statistically significant (*p* < 0.0001) higher concentration of procoagulant (expressing annexin V) microvesicles (MVs) as compared to the control group (Figure 1B). Similarly, a concentration of platelet-derived MVs (expressing CD41 antigen) was significantly elevated in RVO vs. controls (*p* < 0.0284), as shown in Figure 1C. The difference between the RVO group and controls was even more significant (*p* < 0.0001) for a fraction of platelet-derived procoagulant MVs (Figure 1D). Regarding the total population of leukocyte-derived MVs (expressing CD45 antigen), no significant difference was found between RVO patients and controls (Figure 1E); however, leukocyte-derived MVs showed significantly more abundant procoagulant phenotypes in RVO patients vs. controls (Figure 1F). No significant differences were observed between RVO patients and the control group in the cases of both the total population of endothelium-derived MVs (expressing CD144 antigen) (Figure 1G) and endothelium-derived procoagulant MVs (Figure 1H).

We also analyzed differences among various types of MVs, comparing both subgroups of RVO patients (CRVO and BRVO) with controls. We observed that both CRVO and BRVO patients had statistically significant (*p* = 0.0093 and *p* = 0.041 for CRVO vs. controls and BRVO vs. controls, respectively) higher concentrations of procoagulant (expressing annexin V) MVs as compared to the control group (Figure 2A). Interestingly, the concentrations of platelet-derived MVs identified only as CD41-positive objects were not significantly different when CRVO and BRVO patients were compared to controls (Figure 2B); however, the significant differences between both BRVO and CRVO groups and controls were found (*p* = 0.001 and *p* = 0.01 for CRVO vs. controls and BRVO vs. controls, respectively) for a fraction of platelet-derived procoagulant MVs (Figure 2C). Similarly, for a total population of leukocyte-derived MVs (expressing CD45 antigen), no significant differences were observed between CRVO and BRVO patients vs. controls (Figure 2D), whereas the procoagulant fractions of leukocyte-derived MVs were significantly (*p* = 0.01 and *p* = 0.001 for CRVO vs. controls and BRVO vs. controls, respectively) elevated in both CRVO and BRVO subgroups vs. controls (Figure 2E). No significant differences were demonstrated between CRVO and BRVO patients vs. control group for the total populations of both endothelium-derived MVs (expressing CD144 antigen) (Figure 2F) and procoagulant and endothelium-derived MVs (Figure 2G).

### 3.5. Soluble P-Selectin Levels

We assessed the soluble P-selectin (CD62s) concentrations in plasma samples from RVO patients and control group. We found significantly higher values (*p* < 0.0068) of this marker in the RVO group in comparison to controls (Figure 3A). In the cases of CRVO and BRVO subgroups, a statistically significant difference was found for CRVO vs. controls (*p* < 0.0362), but not for BRVO vs. controls (Figure 3B).

## 4. Discussion

Retinal vein occlusion is one of the most common causes among retinal vascular diseases of the worsening of visual acuity [1,13]. Developing complications, such as macular oedema or ocular neovascularization, make RVO a chronic disorder that may lead to a loss of vision [1]. Therefore, from a clinical point of view, finding the causes and markers of this disorder is very important. Currently, however, despite the efforts of many researchers, the pathogenesis of RVO has not been clearly explained and remains multifactorial [26].

It is known that normal hemostasis is dependent on the functioning of platelets as well as vascular and coagulation factors. It is assumed that thrombosis in the retinal venous vessel is a consequence of at least one component of the Virchow’s triad. Despite the fact that the role of blood platelet malfunction in the venous system is not dominant, in the literature there are several papers on the role of platelets in RVO [4]. From a pathophysiological viewpoint, it can be assumed that platelet hyperreactivity could lead to elevated thrombin generation and, consequently, a prothrombotic state [13].

In our work we did not detect differences in any simple platelet parameters obtained from peripheral blood morphology, such as platelet count, MPV, PCT, PDW, and reticulocytes between the RVO group and controls. Notably, the results regarding the significance of these parameters in RVO are inconsistent in the literature. Several reports have found increased MPV values [2,4,17,26,27,29,38], but there is also a study reporting that the value of MPV was significantly decreased in patients with RVO in relation to the control group [13]. On the other hand, in many papers, no association was found [31,34,35,39,40]. A similar inconsistency concerns PDW, PLCR, and total platelet count. Yilmaz et al. reported elevated values of PDW and PLCR in patients with RVO, higher values of the PDW parameter were also noted by Beyazyıldız et al. and Citirik [2,27,38]. Similarly, Kurtul et al. demonstrated no significant statistical association with PDW [35]. Regarding the platelet count, there was only one paper reporting an increased number of platelets in RVO patients [35], while many studies indicated the lack of a significant association [2,4,13,17,26,29,30,31,34,36,37,39]. In addition, other authors emphasized the absence of a relationship with the PCT parameter in RVO [2,39]. In contrast, Beyazyıldız et al. claimed that PCT values were significantly elevated in patients with BRVO [38], while Citirik et al. recorded them in CRVO patients [27]. Overall, therefore, our negative results regarding the significance of these markers are not surprising and appear to agree with many other papers.

Some reports have shown an association between inflammatory markers and the risk of developing RVO. There were papers showing that both systemic and local inflammation can play an essential role in the pathophysiology of RVO [36,41]. One indicator of a systemic inflammatory state is NLR, whose value may be obtained by dividing the number of neutrophils by the lymphocyte count [36]. Another inflammatory marker is PLR (the platelets-to-lymphocytes count) [36]. Our results did not demonstrate any significant association between PLR or NLR and RVO, despite finding some significant differences in individual blood morphology parameters, such as total WBC and its subpopulations (NEU, EOS, BAS) as well as LYM and MON. Notably, as reported in the literature, increased levels of neutrophils were demonstrated in patients with RVO [37] and with BRVO [34,36], which was consistent with our study. In some reports, a significant increase in NLR rate was found in retinal vein occlusion [37], whereas in other works, no association with BRVO [39] or RVO was observed [35]. Similarly, there are papers showing substantially elevated NLR and PLR indexes in BRVO [34,36] or RVO patients [30]. On the other hand, Kurtul et al. concluded that only PLR was significantly increased in RVO patients [35], whereas in another work no significance for either NLR and PLR was found [29], and importantly, the number of patients in this work was remarkably higher in comparison to the previously cited papers.

In our work, we evaluated a number of platelet activation parameters, including platelet aggregation stimulated by ADP, collagen, and arachidonic acid. We did not observe any difference between RVO and controls. In the literature, there are several papers reporting the hyperaggregability of platelets in RVO patients in response to collagen or ADP [14], ADP [18], or thrombin [16]. A possible explanation for the discrepancy between literature data and our results is attributed to the methodology of measuring platelet aggregation. The aforementioned studies used the classical Born’s method described in 1962. Despite this method still being widely used in studies of platelet aggregation, it should be noted that it measures platelet aggregation in platelet-rich plasma (or in suspensions of isolated platelets, as the modification). In our study, we decided to apply a multiple electrical aggregometry method measuring platelet aggregation in whole blood, which is undoubtedly the more natural environment for platelets, resembling bloodstream conditions. Furthermore, our negative results suggesting the lack of differences in the rate of platelet aggregation in RVO vs. controls were supported by our multi-parametric analysis of platelet function using flow cytometry. In this approach, which, as far as we know, was applied for the first time to RVO patients, we evaluated basal platelet activation ex vivo (without stimulation with agonists) and used a model of measuring platelet reactivity in vitro with three platelet agonists, ADP, collagen, and TRAP, acting via different receptors (each agonist was applied in a low and high concentration). Despite using three markers of platelet activation (P-selectin (CD62), PAC-1, and binding of exogenous fibrinogen to platelets) in both models, we did not demonstrate any differences between RVO patients, including CRVO and BRVO subgroups, in comparison with the control group.

Interestingly, we found that the entire group of RVO patients and both BRVO and CRVO subgroups were characterized by a significantly elevated level of the soluble form of P-selectin in plasma (sP-selectin; CD62s). It is well known that under physiological conditions, P-selectin (CD62) is present in the α-granules of platelets [42]; therefore, its expression on the platelet surface is one of the best markers of platelet activation, since a degranulation process is an integral event in the platelet activation cascade. As a result of platelet activation, P-selectin is expressed on the surface of platelets and undergoes a process of rapid shedding [43]. Therefore, a soluble variant of P-selectin present in plasma could serve as another marker of thrombotic events [43,44,45]. P-selectin, thanks to expressing on both platelet and endothelial cell surfaces, plays a significant role in vascular inflammation and the injury process, and is therefore the connection between thrombosis and inflammation [46,47]. In the current literature, only one report has been published indicating the lack of significant differences in the soluble P-selectin level in plasma between patients with retinal vein and artery occlusion and the control group [48], which is contradictory to our results, which unambiguously point to a substantial increase in sP-selectin level in RVO patients. This discrepancy can be explained by the fact that the above cited paper was published almost 25 years ago (1998); therefore, it is difficult to compare completely distinct ELISA tests for sensitivity and specificity. 

In our opinion, the most important finding of our study was the observation that the concentration of procoagulant (expressing phosphatidylserine in the cell membrane, as detected by annexin V binding) microvesicles (MVs) was significantly higher in RVO patients. The changes also remained significant for both CRVO and BRVO subgroups vs. controls. In our experimental approach we were also able to distinguish various types of MVs originating from platelets (expressing the CD41 characteristic exclusively of platelets), leukocytes (CD45 antigen-positive MVs), and endothelium (CD145 antigen-positive MVs). Interestingly, we demonstrated that out of these subpopulations, only platelet-derived MVs were significantly more abundant in the total RVO group as well as in the CRVO and BRVO subgroups. However, for procoagulant MVs, a stronger and significantly higher concentration of MVs was found in RVO, CRVO, and BRVO, for both platelet- and leukocyte-derived MVs, but not for MVs of endothelial origin. In the literature, there are only incidental papers dealing with MV significance in RVO. In general, our results were in accordance with those published by Su et al., who demonstrated that microparticles originating from platelets, erythrocytes, leukocytes, and endothelial cells were elevated in patients with RVO [24]. The only discrepancy regards the role of endothelium MVs. Interestingly, the paper by Su et al. supported the hypothesis that platelet-derived procoagulant MVs could be involved in the procoagulant activity observed in RVO, since they found that among circulating MVs, platelet-derived microparticles predominated in approximately 60–70% [24]. In another study, it was observed that platelet-derived MVs produced in vitro had a procoagulant activity that was 50–100 times greater than that on the membranes of activated platelets [49]. In the light of the above considerations, it is likely that procoagulant activity in RVO patients is largely dependent on circulating platelet-derived MVs and procoagulant MVs from other cells.

The literature has described the role of platelet-derived procoagulant MVs in the progression of atherosclerotic changes in the arterial wall through the influx and adherence of platelets to damaged vascular endothelial sites and the promotion of blood coagulation through providing a procoagulant surface from the abundance of phosphatidylserine in the cell membrane. In addition, platelet-derived microparticles are involved in inflammatory reactions as well as lipid deposition within the atherosclerotic plaque [20]. It can be assumed that the phenomena involving platelet-derived MVs described in the arterial system, especially those associated with the induction of platelet adhesion and enhancing blood coagulation, may also take place in venous vessels, at least partially contributing to the prothrombotic phenotype observed in RVO.

In conclusion, the results of our study suggest that blood platelets can contribute to the etiopathogenesis of retinal vein occlusion by enhancing procoagulant activity in providing a procoagulation surface via platelet-derived microvesicles. The direct role of platelet hyperreactivity in developing RVO is less apparent, which is not surprising, considering the complexity and multifactorial background of this disorder.

### Limitations of the Study

Our study has several limitations. First of all, it is a single-center case control study with relatively low sample size (*n* = 35 for RVO patients). Secondly, we focused our interest on a role of blood platelet function in RVO, however, one should have in mind that this disease is multifactorial. Due to the limited size of the group, it was not possible to apply many factors that could have a potential effect on the results. Moreover, we did not evaluate thrombophilia status in our group. Altogether, our findings, despite in our opinion novel and interesting, should be regarded as rather observational and should be validated in further studies.

## Figures and Tables

**Figure 1 jcm-11-05099-f001:**
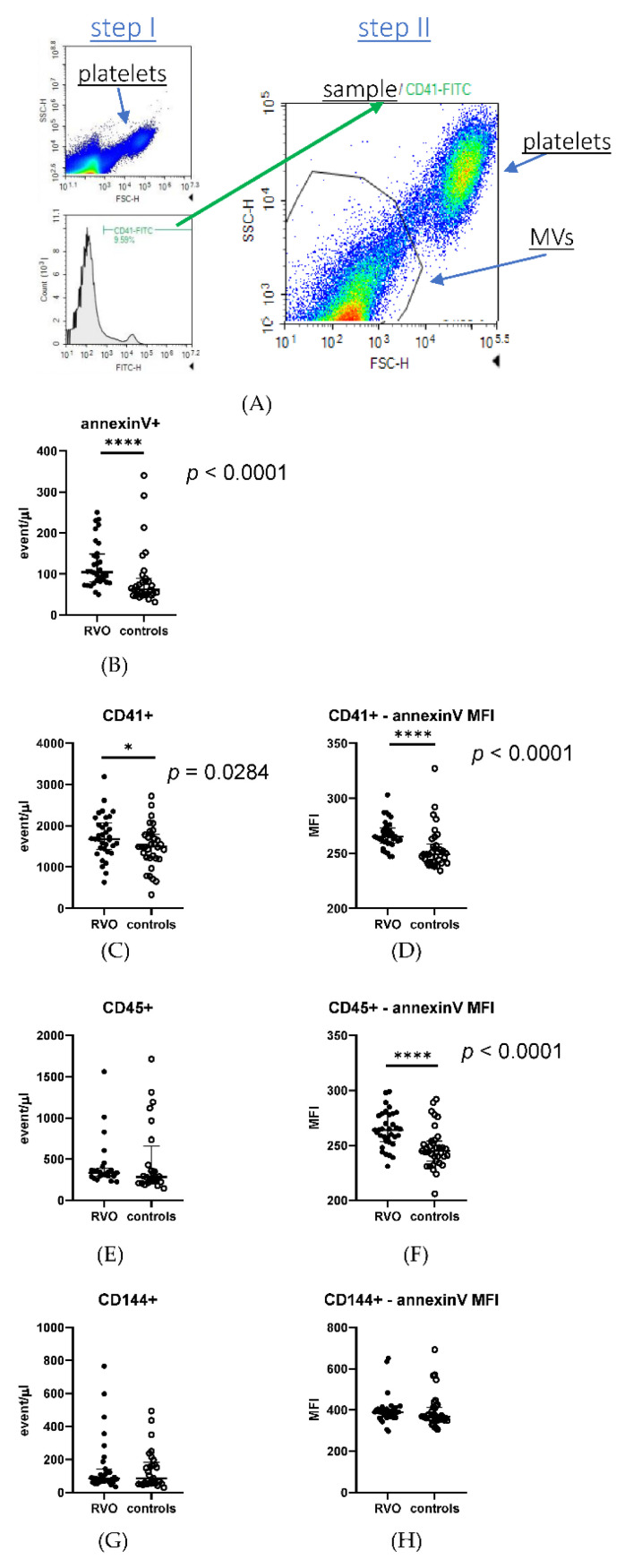
RVO is associated with increased number of procoagulant and platelet-derived microvesicles (MVs). The MV gating schema: the population of antigen-positive events is selected based on a histogram, and within that population the MVs are selected as the events smaller than platelets based on SSC/FSC plot (**A**). The graphs present the dot-plots with median and interquartile ranges (whiskers) of the antigen-positive MV populations, presented as densities, of procoagulant (annexinV-positive) (**B**), platelet-derived (CD41-positve) (**C**), leukocyte-derived (CD45-positive) (**E**), and endothelium-derived (CD144-positive) (**G**) MVs. Graphs (**D**,**F**,**H**) present the annexin V median florescence intensity (MFI) within the indicated antigen-specific MV population. Significance of differences between RVO and controls was estimated using the Mann–Whitney U test. * *p* < 0.05; **** *p* < 0.0001. RVO—Retinal Vein Occlusion.

**Figure 2 jcm-11-05099-f002:**
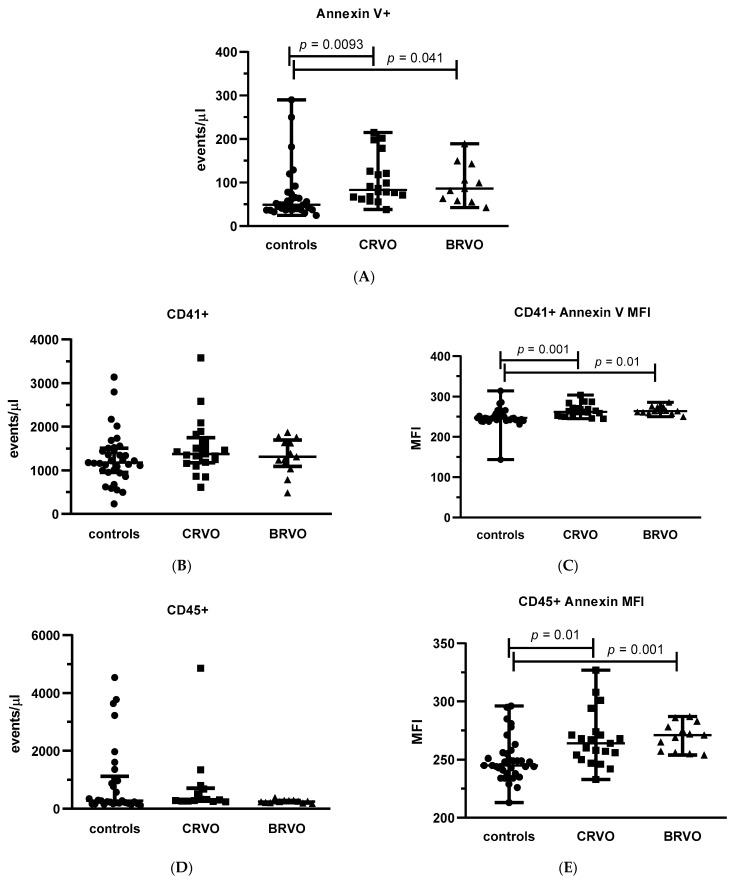
Both subtypes of RVO (CRVO and BRVO) are associated with increased number of procoagulant and platelet-derived microvesicles (MVs). The graphs present the dot-plots with median (horizontal bar) and interquartile range (Q1; Q3) (whiskers) of the antigen-positive MV populations, presented as densities, of procoagulant (annexin V-positive) (**A**), platelet-derived (CD41-positve) (**B**), leukocyte-derived (CD45-positive) (**D**), and endothelium-derived (CD144-positive) (**F**) MVs. Graphs (**C**,**E**,**G**) present the annexin V median florescence intensity (MFI) within the indicated antigen-specific MVs population. Significance of differences between BRVO vs. controls and CRVO vs. controls was estimated using the Kruskal–Wallis non-parametric test with Dunn’s multiple comparison test. BRVO—Branch Retinal Vein Occlusion, CRVO—Central Retinal Vein Occlusion, RVO—Retinal Vein Occlusion. ●—control subjects, ■—CRVO patients, ▲—BRVO patients.

**Figure 3 jcm-11-05099-f003:**
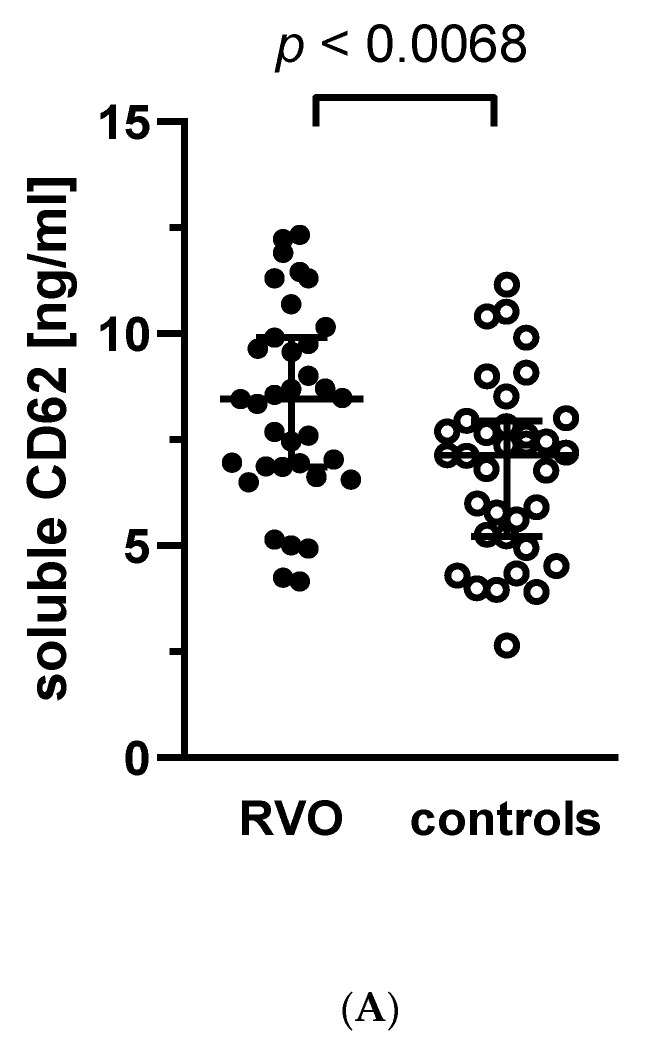
Soluble P-selectin levels (CD62s) are elevated in RVO patients vs. controls (**A**) and in CRVO patients vs. controls (**B**). The graphs present the dot-plots with median (horizontal bar) and interquartile range (Q1; Q3) (whiskers). Significance of differences between RVO and controls was estimated using the Mann–Whitney U test (**A**) and between CRVO or BRVO and controls using one-way analysis of variances (ANOVA) with Bonferroni’s multiple comparisons test (**B**). BRVO—Branch Retinal Vein Occlusion, CRVO—Central Retinal Vein Occlusion, RVO—Retinal Vein Occlusion.

**Table 2 jcm-11-05099-t002:** Basic parameters of peripheral blood morphology in RVO patients and controls.

Parameter	RVO (*n* = 35)	CRVO (*n* = 21)	BRVO (*n* = 13)	Controls (*n* = 35)
RBC (×0^6^) [cells/µL]	4.67 (4.39; 4.86)	4.67 (4.39; 4.86)	4.55 (4.39; 4.86)	4.59 (4.33; 4.91)
WBC (×10^3^) [cells/µL]	6.75 (5.58; 7.69) #	7.49 (6.54; 8.84) #	6.67 (5.33; 6.98)	5.77 (4.98; 6.68)
NEU (×10^3^) [cells/µL]	3.90 (3.10; 4.87) #	4.35 (3.77; 5.22) #	3.5 (2.78; 3.9)	3.30 (2.63; 4.28)
EOS (×10^3^) [cells/µL]	0.18 (0.10; 0.32) #	0.2 (0.12; 0.29)	0.14 (0.10; 0.32)	0.11 (0.08; 0.19)
BAS (×10^3^) [cells/µL]	0.05 (0.04; 0.07) #	0.05 (0.04; 0.06)	0.06 (0.03; 0.07)	0.05 (0.04; 0.07)
LYM (×10^3^) [cells/µL]	2.14 (1.68; 2.46) #	2.18 (1.88; 2.5) #	1.92 (1.66; 2.17)	1.87 (1.56; 2.10)
MON (×10^3^) [cells/µL]	0.59 (0.46; 0.76) #	0.55 (0.49; 0.74) #	0.62 (0.47; 0.77)	0.46 (0.39; 0.52)

Data presented as median and interquartile: Me (Q1; Q3). Statistically significant differences (*p* < 0.05) vs. controls are marked with #. Significance of differences between RVO and control group was estimated using the Mann–Whitney U non-parametric test. Significance of differences between BRVO vs. controls and CRVO vs. controls was estimated using the Kruskal–Wallis non-parametric test with Dunn’s multiple comparison test. RBC—Red Blood Cells, WBC—White Blood Cells, NEU—neutrophils, EOS—eosinophils, BAS—basophils, LYM—lymphocytes, MON—monocytes. BRVO—Branch Retinal Vein Occlusion, CRVO—Central Retinal Vein Occlusion, RVO—Retinal Vein Occlusion.

**Table 3 jcm-11-05099-t003:** Platelet indices calculated from peripheral blood morphology in RVO patients and controls.

Parameter	RVO (*n* = 35)	CRVO (*n* = 21)	BRVO (*n* = 13)	Controls (*n* = 35)
PLT (×10^3^) [cells/µL]	210 (185; 250)	225 (196; 286)	203 (185; 209)	236 (182; 261)
MPV [fl]	11.2 (10.5; 11.7)	11.3 (10.8; 11.7)	10.8 (10.3; 11.7)	10.7 (10.1; 11.9)
PCT (%)	0.24 (0.20; 0.28)	0.25 (0.20; 0.30)	0.22 (0.20; 0.23)	0.25 (0.21; 0.27)
PDW (fl)	13.1 (12.1; 15.2)	13.9 (12.5; 15.3)	12.6 (12.1; 15.2)	12.5 (12.0; 13.4)
Reticulated platelets (×10^3^) [cells/µL]	0.074 (0.052; 0.088)	0.077 (0.060; 0.088)	0.066 (0.05; 0.088)	0.066 (0.056; 0.084)

Data are presented as median and interquartile: Me (Q1; Q3). No statistically significant differences vs. controls were found. Significance of differences between RVO and control group was estimated using the Mann–Whitney U non-parametric test. Significance of differences between BRVO vs. controls and CRVO vs. controls was estimated using the Kruskal–Wallis non-parametric test with Dunn’s multiple comparison test. PLT—platelet count, MPV—Mean Platelet Volume, PCT—plateletcrit, PDW—Platelet Distribution Width. BRVO—Branch Retinal Vein Occlusion, CRVO—Central Retinal Vein Occlusion, RVO—Retinal Vein Occlusion.

**Table 4 jcm-11-05099-t004:** Inflammatory markers in RVO patients and controls.

Parameter	RVO (*n* = 35)	CRVO (*n* = 21)	BRVO (*n* = 13)	Controls (*n* = 35)
PLR	99.1 (82.6; 120.8)	99.1 (78.0; 124.3)	95.7 (93.0; 108.8)	120.6 (82.6; 120.8)
NLR	2.0 (1.5; 2.3)	2.01 (1.5; 2.4)	1.99 (1.5; 2.3)	1.7 (1.4; 2.6)

Data presented as median and interquartile: Me (Q1; Q3). No statistically significant differences vs. controls were found. Significance of differences between RVO and control group was estimated using the Mann–Whitney U non-parametric test. Significance of differences between BRVO vs. controls and CRVO vs. controls was estimated using the Kruskal–Wallis non-parametric test with Dunn’s multiple comparison test. PLR—Platelet to Lymphocytes Ratio, NLR—Neutrophils to Lymphocytes Ratio, BRVO—Branch Retinal Vein Occlusion, CRVO—Central Retinal Vein Occlusion, RVO—Retinal Vein Occlusion.

## Data Availability

The data presented in this study are available on request from the corresponding author. The data are not publicly available in order to protect patients privacy.

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
