# Peer review of "Platelet-Derived Procoagulant Microvesicles Are Elevated in Patients with Retinal Vein Occlusion (RVO)"

_jcm, 2022, doi:10.3390/jcm11175099_

Round 1

Reviewer 1 Report

Platelet-derived indices have significant associations with some conditions such as smoking, hypertension, hypercholesterolemia, obesity, the use of statins and some antihypertensive drugs, etc. Thus, many factors can influence the results.

In the study, the patients in the study group were reported to have hypertension. However, the authors did not mention whether the included patients and controls used antihypertensive medications and statins.

I wonder whether the use of antihypertensive and statin drugs and hypercholesterolemia status were excluded from the study groups. If not excluded I ask the authors to take such parameters into the statistical analysis and report whether these conditions have significant effects on the results.

Reviewer 2 Report

In this manuscript, the authors reported about the relationship between platelet-derived microvesicles, p-selectin and retinal vein occlusion (CRVO and BRVO) in a small single center cohort. Allegedly, this study expands the knowledge on the relationship between RVO and platelet function.

Overall the topic is interesting and the paper is well written although there are some typos that need correction. There are also other issues that I have detailed below:

- Please avoid the use of abbreviations in the abstract - these remains unclear without reading the entire manuscript and do not help the reader to focus on the content of the study.

- The role of hypertension in the pathogenesis of RVO seems totally neglected in the introduction and discussion, although potentially one of the most important factor associated with RVO onset (see for example: https://journals.lww.com/md-journal/Fulltext/2021/10290/Hypertension_as_a_risk_factor_for_retinal_vein.32.aspx). Other cardiometabolic risk factors may be equally important, but are not cited in this maunscript and should be adequately referenced/discussed.

- Relatedly, one may wonder whether the individuals included in this study were tested for thrombophilia. A systematic review and meta-analysis (https://onlinelibrary.wiley.com/doi/full/10.1111/jth.15068) have depicted the epidemiology of thrombophilia in RVO patients; this is particularly important in younger patients, which may have less other risk factors for RVO. Given the mean age of the patients included, one may wonder whether thrombophilic patients were included/thrombophilia assessed. This could have influenced the results. If thrombophilia was not assessed, this should be listed in the limitation and appropriate discussion made.

- Similarly, can the authors specify if they have excluded also patients taking anticoagulants? This is important since several anticoagulants can influence platelet function.

- Also, can the authors specify whether the blood draws were done before or after any treatment for RVO was started? This is important since steroids and, more specifically, anticoagulants are currently used in the treatment of RVO and may have altered the results presented.

- Please report full p-values - simply reporting p<0.05 is not enough and should be avoided. This refers to both text and tables

- As for the tables, what are the significance levels standing for? In other words, to which comparisons these p-values refer? Please specify.

- The biological rationale behind the association between platelet MV and the risk of RVO should be explained more in the introduction - why are the data observed potentially important? this also apply to the discussion, in which the authors should go beyond listing previous findings on the topic, and rather discuss what are the novel implications of their study.

- One should note that, according to the relatively low sample size, and the potential for unaccounted confounders (which are many, and not accounted in the analysis plan - this should be listed as limitation), the findings presented are merely descriptive and rather speculative. this should be reflected more in the discussion. Note that alle the analyses were unadjusted for potential confounders.

Round 2

Reviewer 1 Report

The revised version is enough for me. The paper was improved.

I have no more comments.

Thank you.

Reviewer 2 Report

No further comments